# “Noncovalent Interaction”: A Chemical Misnomer That Inhibits Proper Understanding of Hydrogen Bonding, Rotation Barriers, and Other Topics

**DOI:** 10.3390/molecules28093776

**Published:** 2023-04-27

**Authors:** Frank Weinhold

**Affiliations:** Theoretical Chemistry Institute and Department of Chemistry, University of Wisconsin-Madison, Madison, WI 53706, USA; frank.weinhold@wisc.edu

**Keywords:** natural resonance theory, bond order, hydrogen bonding, rotation barrier, steric repulsion, natural bond orbital analysis, supramolecular interaction, σ/π-hole theory

## Abstract

We discuss the problematic terminology of “noncovalent interactions” as commonly applied to hydrogen bonds, rotation barriers, steric repulsions, and other stereoelectronic phenomena. Although categorization as “noncovalent” seems to justify classical-type pedagogical rationalizations, we show that these phenomena are irreducible corollaries of the *same* orbital-level conceptions of electronic covalency and resonance that govern *all* chemical bonding phenomena. Retention of such nomenclature is pedagogically misleading in supporting superficial dipole–dipole and related “simple, neat, and wrong” conceptions as well as perpetuating inappropriate bifurcation of the introductory chemistry curriculum into distinct “covalent” vs. “noncovalent” modules. If retained at all, the line of dichotomization between “covalent” and “noncovalent” interaction should be re-drawn beyond the range of quantal exchange effects (roughly, at the contact boundary of empirical van der Waals radii) to better *unify* the pedagogy of molecular and supramolecular bonding phenomena.

## 1. Introduction

The history of chemistry is marked by the watershed shift that accompanied the earliest glimmerings of the covalency concept in the mid-19th century [1]. The mysterious linkages of atomic valencies that came to be identified as “covalent” bonds were the centerpiece of Kekulé’s structural theory [2] that initiated the rapid growth of organic chemistry and dominated subsequent developments in chemical industry, research, and pedagogy. It is remarkable that this key concept, including its elaborations to the conjugative and resonance-type phenomena of aromatic species, achieved a high degree of practical usage *before* underlying notions involving electrons, orbitals, and their quantal machinations came to recognition.

Kekulé’s theory was originally cloaked in somewhat murky symbolism and nomenclature that barely concealed its clear violations of then-accepted precepts of Newtonian mechanics. However, Kekulé’s insights were vindicated in all subsequent developments, including Lewis’s electron-dot formulation of bonding and acid/base character (1923) [3], Schrödinger’s discovery of the wave equation governing electronic behavior (1926) [4], and Pauling’s masterful exposition of bonding principles in terms of qualitative hybridization and resonance concepts (1931,1939) [5,6].

As quantitative computer-based solutions of the polyatomic Schrödinger equation began to appear in the 1960s and 1970s, chemistry textbooks and department organization continued to reflect bifurcation into “covalent” vs. “noncovalent” aspects of chemical theory. Such dichotomization is also recognizable in textbook treatments of molecular vs. intermolecular interactions, chemical vs. physical forces, covalent vs. ionic bonding, and organic vs. inorganic reasoning. In retrospect, many such dichotomizations appear illusory, anachronistic, and ultimately unsustainable in the face of *unified* quantum-chemical advances in all branches of the chemical, biological, and material sciences.

In the present work, we wish to demonstrate that key concepts commonly relegated to the “noncovalent” domain—hydrogen bonding, rotation barriers, and steric repulsions—should instead be recognized (and taught!) as inseparable aspects of general orbital-level covalency. The problem is fully evident in the research literature as well as textbooks, but efforts to address the problem should begin at a didactic level with minimal presumption of mathematical background. In this manner we wish to add support to recent criticisms of common textbook rationalizations in terms of quasi-classical dipole–dipole conceptions, which are aptly described as “neat, simple, and wrong” [7].

The results to be presented below depend principally on a powerful “trick” of natural bond orbital (NBO) analysis [8] in the framework of density functional theory (DFT) calculations [9]. The trick consists of *deleting* specific NBOs or their interactions and *recalculating* the DFT potential energy surface [10] (including a new equilibrium geometry, energetics, vibrational frequencies, etc.) as though these orbitals or interactions were absent in nature. In this manner, one can track the appearance or disappearance of a particular phenomenon—such as a hydrogen bond or rotation barrier—to the specific “smoking gun” NBO interaction that is implicated as a unique cause of the phenomenon. A mere glance at the *difference* in calculated properties in the full (*E*) vs. deletion (*E*^(DEL)^) calculation allows causality to be inferred for the phenomenon of interest.

According to well-known pedagogical precepts, we should try the simplest cases first. For H-bonding, elementary diatomic hydrides are the candidates of simplest “dipolar” form, allowing the most direct possible comparison of orbital-level covalency vs. quasi-classical dipole–dipole rationalizations of their intermolecular interactions. In such comparisons, it is advantageous to focus on qualitative differences in supramolecular *shape* or other properties that have counterparts in everyday experience, relieving the dependence on mathematical details that beginning students may find challenging. Ideally, each alleged “noncovalent” property should be discussed in the selfsame module where orbital-level details of chemical bonding and periodicity are first confronted in the chemical curriculum. Indeed, everyday experience with “tricky dog magnets” (which Google) may adequately represent student intuitions concerning classical dipole–dipole conceptions, which can then be compared with NBO-based orbital shapes and their intuitive dependence on periodic electronegativity trends that are the essence of quantum chemistry description. Increasingly, both introductory students and accomplished chemical practitioners have access (e.g., through *WebMO* [11] or other programs described below) to quantum chemical results and graphical orbital imagery that allow them to replace the elementary examples presented here with species of their own choosing.

## 2. Computational Methods

For all the examples presented below, we employed the routine B3LYP/6311++G** level of density functional theory. However, the qualitative conclusions drawn here are largely insensitive to details of the DFT functional or basis set, so long as the latter includes diffuse functions (“++” augmentation) to describe the longer range of interactions in the H-bonding regime. The present results were obtained with the *Gaussian 16* [12] host electronic structure system interfaced with the *NBO 7.0* [13,14,15] analysis program, and orbital imagery was generated with the *NBOPro7@Jmol* [16] utility program. Associated natural resonance theory (NRT) bond orders [17] were obtained with the NRT keyword module of *NBO 7.0.* The run-ready input files provided in the Appendix A make it easy to obtain comparable results for virtually any flavor of DFT in current usage.

## 3. Orbital-Level Covalency of Hydrogen Bonding

### 3.1. H-Bonding in Diatomic Hydrides

As simplest cases of AH···B hydrogen bonding, we consider the self- and cross-interactions of elementary hydrogen halides (XH, X = F, Cl, Br) and the related (X = H) diatomic H_2_ that *lacks* a dipole moment and corresponds to the weakest limit of recognizable H-bonding. Figure 1 shows the optimized equilibrium geometry (and parenthesized net binding energy in kcal/mol) for all twelve possible supramolecular complexes from these precursors. All these complexes are seen to exhibit severely bent (L-shaped) geometry, with more-or-less linear X-H···Y alignment along the long axis between two halogens, *the characteristic structural signatures of H-bonding*.

Other features of this family of complexes are immediately evident. From the strong (near-perpendicular) deviations from overall linear geometry, one also sees that these complexes deviate *as strongly as possible* from the expected doggie-magnet alignment geometry of classical dipole–dipole interactions. Moreover, with the exception of X = Y = H, each XH···YH complex shown in the “XH” row and “YH” column of Figure 1 is complemented by a *related* YH···XH complex shown in the “YH” row and “XH” column. Thus, each diatomic XH monomer has the ability (even with near-*equal* propensity) to serve either as the H-donor or H-acceptor in the H-bonding geometry. This indicates that we are not looking at “competition for the H-atom” (as conventional terminology might suggest) but rather that the H atom serves to “bridge” the resonance-type bonding character with one terminal halogen or the other, viz. (in conventional resonance structural symbolism),
X–H···Y ↔ X···H–Y(1)
Indeed, if we include an additional bond on each side of the resonance arrow, we can see obvious resemblance to, e.g., the resonance-type representation of the peptide group by a protein chemist, viz.,
O=C–N ↔ O–C=N(2a)
or the analogous representations of allyl-type resonance by an organic chemist, viz.,
C=C–C ↔ C–C=C(2b)
In each case, what is being “donated” or “accepted” is not a hydrogen atom but rather a *shared electron-pair bond*, the essential change of perspective from original Brønsted–Lowry to the more prescient Lewis formulation of acid–base interactions [3]. All such “covalency”-type conceptions appear deeply in conflict with classical electrostatics. The *electrons* (not the protons) are the dominant “quantum particles” in chemistry.

As we learn from Bohr theory, description of electronic phenomena must always involve the available *orbitals* in which electrons may be “found” (with probability distribution as detected experimentally). In the case of a diatomic hydride HX, the bonding orbital for a shared *e*-pair (denoted σ_HX_) may be expressed as the *in*-phase linear combination of hybrid atomic orbitals (denoted *h*_H_, *h*_X_), viz.,
σ_HX_ = *c*_H_ *h*_H_ + *c*_X_ *h*_X_(3a)
where
|*c*_H_|^2^ + |*c*_X_|^2^ = 1(3b)
The linear combination (Equation (3a)) expresses the superposition [18] of atomic orbitals *h*_H_, *h*_X_ to form a new “bond orbital” σ_HX_, thereby putting into mathematical form what the chemist describes as the “sharing” (equal or unequal, according to values of *c*_H_, *c*_X_) of the *e*-pair between atoms—the essence of covalent bonding.

Equation (3b) also expresses an important *conservation* principle of orbital construction. In a certain sense, each starting hybrid atomic orbital *h*_H_, *h*_X_ in Equation (3a) is like an axis of a many-dimensional Cartesian space, and the coefficients *c*_H_, *c*_X_ serve as components of the σ_HX_ orbital in each spatial direction. Whether the σ_HX_ orbital is “tilted” more into the *h*_H_ or the *h*_X_ direction, the sum of squared components (Equation (3b)) must always remain *constant* (the orbital remains “normalized”). Since the probability of detecting an electron “in” σ_HX_ (i.e., in the spatial region where σ_HX_ has finite amplitude) is related to the square of the orbital amplitude (|σ_HX_|^2^), the conservation condition (Equation (3b)) guarantees that the total probability for finding the *e*-pair in σ_HX_ is independent of its equal or unequal sharing between *h*_H_, *h*_X_ hybrids. This merely assures the unitary condition that the total number of electrons undergoes neither gain nor loss in the domain of chemical transformations.

The specific forms of natural hybrid orbitals (NHOs) *h*_H_, *h*_X_ and the resultant bonding NBO σ_HX_ are readily obtained from NBO analysis output of the quantum chemical calculation for each HX. Figure 2 displays the 3D shapes of these orbitals and the composition (Equation (3a)) of each bonding σ_HX_ in the left three panels of each row. The chosen orbital surface contour approximates the “van der Waals contact” limit (discussed below) at which interatomic interactions become comparable to ambient thermal energies. The exponentially steep decline of orbital density beyond this limit (as depicted in a following section) ensures that chemically significant NBO interactions remain essentially *localized* (2- or 3-center) and highly recognizable rather than “completely delocalized” as suggested by the florid graphical imagery of canonical molecular orbitals. The latter are merely one of *infinitely* many alternative ways of choosing idealized doubly occupied orbital “axes” to describe a DFT calculation (with no effect on the density or other measurable property), thereby disguising the nuances of localized electronic-pair sharing between *proximal* atomic centers that is the essence of covalent bonding.

The σ_HX_ bonding orbitals shown in the third column of Figure 2 might seem to be the end of the covalency story, but they are not. As mentioned near Equation (3a), each component of an orbital superposition acts like an independent axis of a Cartesian coordinate system and the coefficients of each superposition satisfy a conservation relation (3b), ensuring that the resultant orbital also preserves such axis-like character in a *rotated* Cartesian axis system. This can be made explicit by formally defining a rotation angle (θ_HX_) such that *c*_H_ ≡ cos(θ_HX_) and *c*_X_ ≡ sin(θ_HX_). Equation (3a,b) can then be rewritten as
σ_HX_ = *h*_H_·cos(θ_HX_) + *h*_X_·sin(θ_HX_)(4a)
sin^2^(θ_HX_) + cos^2^(θ_HX_) = 1(4b)
Equation (4b) is a well-known trigonometric identity that is true for *all* rotation angles θ_HX_ (all possible physical values of bond polarity) so that the conservation property (Equation (3b)) is automatically maintained in such a geometrical picture of the superposition. However, the elementary geometry of rigid-body rotations tells us that Equation (4a) can represent only *one* axis of the rotated Cartesian system, whereas the second axis (which ensures conservation of dimensionality under rotations) must be the corresponding *out*-of-phase superposition, denoted σ*_HX_ (“antibonding”), viz.,
σ*_HX_ = *h*_H_·sin(θ_HX_) − *h*_X_·cos(θ_HX_) = *c*_X_·*h*_H_ − *c*_H_·*h*_X_(4c)
A little vector algebra then establishes that if orbitals *h*_H_, *h*_X_ are mutually perpendicular (such as the axes of a Cartesian system), then so are the orbitals σ_HX_ and σ*_HX_. In effect, the σ*_HX_ orbital must *always* be created in synchrony with the σ_HX_ orbital, “completing the space” of the orbital superposition. The inexorable logic of quantum theory (based on the Cartesian-like qualities of many-dimensional Hilbert space) demands that σ_HX_ and σ*_HX_ are inextricably linked in the domain of covalency.

But what can be the physical meaning of such an *anti*bonding (out-of-phase) NBO? To answer this, we recall one other aspect of quantum covalency, namely, the *Pauli exclusion principle*, a consequence of the exchange antisymmetry property of many-electron wavefunctions. This principle restricts the maximum electronic occupancy of any orbital to one *pair* of electrons: one with spin angular momentum “up” and the other with spin angular momentum “down”. However, the complementary σ*_HX_ orbital provides the *capacity for change* in electronic bond shifts, i.e., for receiving the electronic charge “transfer” (CT) from one closed-shell molecule to another (or from another filled bond orbital in the same molecule). Analogous to atoms of the periodic table, which remain chemically *inert* unless one or more valence-shell orbitals are vacant, so does a diatomic HX molecule require a σ*_HX_ “acceptor” NBO to enjoy the benefits of CT-type bond–antibond interactions with a doubly occupied “donor” NBO of a nearby HY molecule. As shown by a simple second-order perturbation theory argument [18,19], each such “donor–acceptor” interaction is intrinsically *stabilizing* because its magnitude (*E*_d-a_^(2)^) depends only on the *square* (|***h***_d,a_|^2^) of the ***h***_d,a_ transition amplitude and the non-negative energy difference (ε_a_–ε_d_) between acceptor and donor NBOs no matter what combination of kinetic and potential energy contributes to the total energy of the system.

NBO analysis provides a read-out of leading donor–acceptor interactions for each XH···YH complex. Invariably, the leading acceptor orbital is the σ*_HX_ valence antibond (“BD*” in NBO output) of the HX monomer that lies along the H-bond axis. The leading donor orbital is generally an off-axis lone pair *n*_Y_ (“LP” in NBO output, generally of high p-character) on the YH monomer. Each such *n*_Y_- σ*_HX_ interaction leads to the corresponding second-order perturbative estimate of stabilization (*E*_n-σ*_^(2)^). Figure 3 depicts the leading *n*_Y_ -σ*_HX_ donor–acceptor interaction (and parenthesized *E*_n-σ*_^(2)^ estimate; kcal/mol) for each HX···YH complex in Figure 1.

The high transferability [20] of NBOs and consistency of *n*_Y_-σ*_HX_ near-alignment in XH···YH complexes is evident throughout Figure 3. Overall, one can see that *all* these clusters prefer the L-shaped geometry that is dictated by *n*_Y_-σ*_HX_ co-alignment, which is directly opposed to any plausible “dipole–dipole” logic. The left-hand column of Figure 3 shows that HF generally prefers a slightly more open angle between the two arms of the L-shape (perhaps reflecting a residual dipole–dipole tendency toward overall linearity), but the strong dominance of *n*_Y_-σ*_HX_ orbital interaction over any presumed dipole–dipole driving force of H-bonding is evident throughout the series.

Additional questions may remain about the small deviations from idealized linearity of H-bonding, as reflected in the apparent failure of *n*_Y_ and σ*_HX_ NBOs to achieve the perfectly collinear “maximum overlap” geometry that their shapes suggest. Other questions concern the differences between *E*_n-σ*_^(2)^ estimates (Figure 3) vs. the actual net binding energies Δ*E* (Figure 1) of the complexes. Both types of deviations largely reflect the role of *steric repulsions*, which will be discussed below as an integral feature of the total quantum covalency picture.

### 3.2. NBO-Deletions Analysis of H-Bonding

In the former section we argued from first principles that bonding (σ_HX_) and antibonding (σ*_HX_) orbitals are inseparable aspects of covalent bonding, and the latter are *unique* prerequisites for the *n*_Y_-σ*_HX_ donor–acceptor interactions that consistently appear in NBO analysis of H-bonding in the studied XH···YH complexes (as well as all other known H-bonded species). The species chosen for study generally lie toward the weak limit of H-bonding, in contrast to a previous study [21] where progressively stronger H-bonds were found whose NRT bond orders ranged continuously upward toward full integer values.

The consistent importance of charge transfer (CT) from donor *n*_Y_ to acceptor σ*_HX_ NBOs in H-bonded X–H···:Y species draws a sharp distinction between NBO-based and “HOMO-LUMO”-based analyses of supramolecular interactions. The σ*_HX_ NBO (valence antibond, “BD*” in NBO output) may superficially appear equivalent to a “virtual” orbital in MO theory or, more specifically, to the lowest unoccupied (LUMO) orbital of canonical SCF-MO or DFT calculations. By definition, however, any virtual MO has *zero* occupancy and makes *no* contribution to H-bonding or any other observable property. Similarly, for multi-center molecules of appreciable complexity, the highest-energy doubly occupied molecular orbital (HOMO) often differs unrecognizably from the relevant (proximal!) *n*_Y_ NBO of the critical *n*_Y_-σ*_HX_ donor–acceptor interaction. The CMO keyword [22] of NBO analysis provides quantitative details of the relationship between NBOs and MOs that allows any confusion between HOMO-LUMO and *n*_Y_-σ*_HX_ involvement in X–H···:Y hydrogen bonding to be quickly resolved.

A more direct and dramatic demonstration of the *necessity* for σ*_HX_ involvement in H-bonding can be given for the current data set of XH···YH complexes. For any MO/DFT method (with well-defined Fock/Kohn-Sham 1-electron energy operator), the NBO program allows the user (with $DEL…$END keylist input [23]) to *delete* one or more low-occupancy (“non-Lewis”) NBOs (or any of their specific donor–acceptor interactions) and *recalculate* the new (*E*^(DEL)^) potential energy surface and its altered equilibrium geometry, vibrational, and reactive features. Deleting the σ*_HX_ NBO removes the capacity for *n*_Y_-σ*_HX_ stabilization and thereby *raises* the variational energy (*E*^(DEL)^ > *E*^(full)^) by an amount that approximates the corresponding *E*_n-σ*_^(2)^ perturbative estimate. However, the variational *E*^(DEL)^ recalculation also allows one to quantify the many additional structural and vibrational effects of σ*_HX_ participation in terms of associated *differences* between *E*^(DEL)^ and *E*^(full)^ potential energy features. For the present XH···YH complexes, we specifically removed *both* σ*_HX_ and σ*_HY_ NBOs because removing only σ*_HX_ would merely cause the XH···YH complex to rearrange to the alternative YH···XH coordination.

Figure 4 displays the reoptimized *E*^(DEL)^ structure in a way that allows direct comparisons with the corresponding panels of Figure 1. One can see that the *E*^(DEL)^ structures differ unrecognizably from the orderly H-bond patterns of Figure 1. Even where some distant similarity remains to the *E*^(full)^ structure of Figure 1 (as, e.g., for FH···FH or ClH···BrH), the inter-monomer separation is significantly increased (up to or beyond the formal distance of van der Waals contact), the net binding energy is significantly reduced (often to the much weaker range of dispersion and other near-ambient thermal effects), and the characteristic structural signatures of H-bonding are *absent* (with *R*_F(3)H(4)_ *shorter* than *R*_F(1)H(2)_ and so forth). In short, one concludes that removal of σ*_HX_ and σ*_HY_ NBOs *annihilates* recognizable H-bonding features in these complexes. Logically, this serves as definitive proof that *presence* of the hydride antibonds is a *necessary* condition for realistic H-bonding.

More specific deletions of individual *n*_Y_-σ*_HX_ matrix elements further confirm the *unique* dependence of realistic H-bond properties on this particular donor–acceptor interaction in all XH···YH complexes. In this connection, it should also be noted that deletion of the σ*_HX_ NBO has *no* effect on any property of the isolated HX monomer, because the numerical occupancy of the σ*_HX_ NBO is automatically *vanishing* in any such diatomic species. Thus, the *E*^(DEL)^ calculation can be said to *exactly* preserve the dipole moments and other steric and electrostatic properties of isolated monomers but with *failure* to preserve realistic vestiges of H-bonding.

### 3.3. Natural Resonance Theory (NRT) Analysis of H-Bonding

A far simpler and chemically intuitive description of H-bonding is provided by NRT *bond orders* {*b*_AB_} for each atom pair in the complexes of Figure 1. Beginning students are commonly introduced to the elementary single, double, and triple bond orders (and resonance-type fractional intermediates) of simple organic species, and intermediate students learn of the higher bond orders that are common in metal–metal bonding [24]. However, inadequate attention is given to the *supra*molecular bond orders of hydrogen bonding (and analogous halogen bonding, pnictogen bonding, etc.) that merely extend the continuous range of resonance-covalency bonding motifs one notch further downward into the *sub*-integer range.

Figure 5 presents the calculated NRT bond orders for all near-neighbor atom pairs in the twelve XH···YH complexes, allowing direct comparisons with the structural data of Figure 1. The intramolecular bond order value (*b*_XH_) for each Lewis acid monomer XH (except HH, where the weak resonance effects lie below the default NRT search threshold) is seen to exhibit the slight reduction (commensurate with corresponding slight elongation in Figure 1). The intermolecular *b*_H···Y_ bond orders also exhibit the expected increase with reduced *R*_H···Y_ separation (and increased H-bond strength) that is expected on intuitive grounds from general bond order–bond length (BOBL) correlations. The correlations of NRT bond orders with various experimental and theoretical descriptors of H-bonding are further quantified in the following section.

### 3.4. Consistent Correlations with Experimental Signatures of H-Bonding

Do the structural features exhibited in Figure 1 (and annihilated by *E*^(DEL)^ deletions in Figure 4) constitute authentic H-bonding? Although the answer seems obvious “by inspection”, we can apply the operational criteria as recently adopted in the IUPAC *Gold Book* definition of the hydrogen bond [25] to address the question. Similar criteria were applied in previous studies [26] of much stronger H-bonds, including bifluoride anions (FHF^−^) and Zundel cations (H_5_O_2_^+^) with binding energies roughly an order of magnitude greater than those of the present study.

In short, the IUPAC definition identifies well-established experimental signatures of XH···Y H-bonding (such as ν_HX_ vibrational red-shifting, *R*_HX_ covalent bond elongation, sub-van der Waals *R*_H···Y_ approach distance, anomalously high proton-NMR shielding, etc.) and requires that as many of these criteria as possible be verified for their consistent correlations with trends exhibited by consensus examples of H-bonding. Theoretical descriptors of H-bonding (such as NRT bond orders *b*_HX_, *b*_H···Y_) or new experimental technologies can similarly be tested for mutually consistent correlations with established experimental criteria of H-bonding, thereby building a more stringent and comprehensive characterization of H-bonding as experimental and theoretical methodologies progress.

In particular, inclusion of NRT bond orders makes explicit the deep connection of H-bonding to resonance covalency, confirming the apparent parallels noted in Equations (1) and (2a,b) above. The present species all lie toward the lower end of recognizable H-bonding, where BOBL correlations must eventually *deviate* from linearity (because *R*_H···Y_ finally increases without limit as *b*_H···Y_ → 0). The idea of *fractional* bond orders (particularly in the sub-integer range) may seem puzzling when first introduced, but such *continuously variable* descriptors of chemical bonding merely reflect the ubiquity of resonance-type phenomena throughout the chemical domain, further extending the obligatory invocation of resonance in discussions of aromaticity or peptide chemistry. Chemical teaching based on the perception of disjoint “covalent” and “noncovalent” domains serves only to obscure the broader role of resonance-type superposition [18]. Such pedagogical misconceptions must eventually yield to current rapid advances in understanding supramolecular phenomena [27,28].

Table 1 summarizes calculated values for a variety of experimentally measurable and theoretical descriptors of H-bonding in the XH···YH complexes of Figure 1, ordered by net binding energy (Δ*E*_HB_, kcal/mol) in column one. The successive columns present numerical values for the shift in IR stretching frequency (Δν_XH_, cm^−1^), bond length (Δ*R*_XH_, Å), and NRT bond order (Δ*b*_XH_) of the covalent HX bond, followed by the net charge transfer (*Q*_CT_, *e*) and perturbative estimate of *n*_Y_-σ*_HX_ stabilization (*E*_nσ*_^(2)^, kcal/mol) in complex formation. These descriptors still neglect steric effects on net binding energy Δ*E*_HB_, but their ordering according to this property allows one to directly see expected types of correlations with other properties.

Correlations between experimental and theoretical descriptors of H-bonding are exhibited more explicitly in Table 2, which presents values of the Pearson correlation coefficient (χ) for each pair of descriptors in Table 1. The χ-values exhibit the expected sign of the correlation in all cases. The magnitude |χ| is in the range 0.6–0.9 for correlations with net Δ*E*_HB_, but it is significantly stronger (0.8 or higher) among the various measurable descriptors of the acceptor HX monomer or NBO/NRT characterizations of *n*_Y_-σ*_HX_ interactions. All these correlations suggest that the experimentally measurable (Δν_XH_, Δ*R*_XH_) and NBO/NRT-based (Δ*b*_AH_, *Q*_CT_, *E*_nσ*_^(2)^) descriptors of H-bonding are mutually complementary and consistent for the entire data set of XH···YH complexes, including the case X = H where “dipole” character is absent in the XH acceptor (Lewis acid) monomer.

The foregoing pedagogical examples were chosen for clarity and simplicity, but analogous results are reliably predictable for *any* H-bonded species of the reader’s choice. All such results focus attention on the importance of σ*_HX_ valence antibond orbitals as the “missing link” (Equation (4c)) of covalent bond formation that ties H-bonding (and other supramolecular complexation phenomena) to the same basic quantum superposition principles that underlie other aspects of covalency throughout the molecular sciences. Merely introducing σ*_HX_ as the necessary complement of σ_HX_ bond formation (at the same time the latter is introduced) can smoothly provide the firm foundation for the later introduction to H-bonding, sparing students the (ultimately futile) digressions into classical multipole electrostatics.

As the name implies, the “non-Lewis” orbitals represent the *capacity for change* from the basic Lewis-structural bonding pattern to alternative resonance-structural patterns, which is the essence of resonance conceptions in chemistry. Expected complementary shifts in the *shapes* of bonding σ_HX_ and antibonding σ*_HX_ NBOs with changes in polarity are expected to *modulate* the primary *n*_Y_-σ*_HX_ interaction, similar to the manner in which heteroatomic substitutions shift the nuances of resonance-type aromaticity, but the primary focus of introductory chemical pedagogy should be to emphasize the *covalent* aspects of H-bonding, deprecating usage of “noncovalent” terminology.

### 3.5. “Sigma-Hole” Picture of H/X-Bonding

A Reviewer has requested consideration of alternative “σ-Hole” nomenclature and conceptions of H-bonded and general X-ogen (X = halogen, pnictogen, etc.) bonded species [29,30,31,32]. This approach builds on the use of the electrostatic potential (ESP) V(**r**),
V(**r**) = ∑_A_ Z_A_/|**R**_A_ − **r**| − ∫ [ρ(**r**′)/|**r**′ − **r**|] d**r**′(5)
whose value at a chosen point **r** (e.g., **r**_LB_ of a nearby Lewis base) depends on the locations (**R_A_**) of charged nuclei (Z_A_) and the Coulombically averaged electron density (ρ) of the chosen molecule (e.g., a hydridic Lewis acid). The strength and sign of a Lewis acid–base interaction is thereby considered to be electrostatically driven [30] if a point (**r**_σ_) of reduced electron density (“σ-hole”) near the hydride terminus leads to V(**r**_σ_) of *opposite* sign (in the “intuitive” classical picture of Coulombic attraction) to that of the corresponding V(**r**_LB_) of the Lewis base. The **r**_σ_ depletion point is chosen along a particular contour (0.001 a.u.) for rendering the V(**r**) surface plot in color-coded gradations, leading to the colorful “bulls-eye” graphical images of σ-holes that are commonly featured in ESP-based analyses.

The “σ-hole” terminology appears less biased than “noncovalent” for expressing its relationship to covalency conceptions as taught elsewhere in the chemistry curriculum. However, the theoretical foundation for ESP-based interpretations is claimed to be the Hellmann–Feynman theorem, as paraphrased in a form (“The force exerted upon any nucleus in a molecule is entirely classically Coulombic” [30]) that leads to still stronger denial of the role of resonance covalency in supramolecular bonding (e.g., “*There is no need to invoke charge transfer*” [29]). Stated in summary form, it is claimed that the ESP-based framework provides “quick and easy qualitative interpretation” [29] and “supports the interpretation of these σ-, π-hole interactions as Coulombic in nature, which is consistent with the rigorous Hellmann-Feynman theorem” [31].

However, the rigor of the Hellmann–Feynman theorem is two-edged. The formal theorem is known to be valid if (and only if) expressed in terms of the *exact* electron density ρ, which in turn must be strictly related to the exact quantum-mechanical wavefunction (unobtainable from classical electrostatics alone) [33]. Unlike other theorems of quantum chemistry, the Hellmann–Feynman theorem has *no* underlying stationary or variational properties to guarantee its stability with respect to small errors in the assumed form of ρ. Indeed, it was recognized long ago that apparently insignificant errors in ρ(**r**) can lead to grossly *un*physical Hellmann–Feynman predictions of physical properties [34].

In practice, ESP-based description of intermolecular binding energies is based not on the Hellmann–Feynman theorem or other theorems of quantum chemistry but rather on *correlations* (regression fits) for varying numbers of “electrostatics-related” properties [beyond basic V(**r**_σ_), V(**r**_LB_) values]. As statisticians often remind us, correlation is not causation. The presented correlations [31,32] typically involve different numbers, identities, and numerical coefficients in fits for different data sets of Lewis acid–base interactions. Examples include a four-term fit [with added dependence on electric field ℇ(R) and dipole polarizability α values] and a five-term fit [with added dependence on V(R) along the line R of acid–base separation] for tabulated σ-hole interactions but a *different* four-term fit for π-hole interactions and still other fits for other data sets in which basic V(**r**_σ_) or V(**r**_LB_) dependence is omitted [31]. Despite their relatively unconstrained forms, the best such fits are marked by significant “outlier” error (ca. 8 kcal/mol, 36% of Δ*E*) that is attributed to neglected contributions of “secondary interactions”.

All such features of ESP-based interpretations of H/X-bonding may be compared with corresponding NBO/NRT-based graphical and numerical descriptors of resonance-type charge-transfer interactions in Section 3.1, Section 3.2, Section 3.3 and Section 3.4. The latter appear in consistent and unified fashion in *all* known H-bonded and X-bonded species. Indeed, if we consider the changes in shape of the hydride σ*_AX_ NBO (automatically coupled to those of the corresponding σ_AX_ NBO) under changes of polarization coefficients *c*_A_, *c*_X_ [cf. Equation (4a–c)], the similarity to the changes in electron density polarized by a unit charge near the sigma-hole becomes apparent (Figure 2 of Ref. [29]). Such similarities assure that the same NBO/NRT picture of resonance covalency will apply consistently to all the species mentioned in Refs. [29,30,31,32].

## 4. Orbital-Level Covalency of Rotation Barriers and Related Stereoelectronic Phenomena

### 4.1. Role of Valence Antibonds in Torsional Barriers

Similar misusage of “noncovalent” terminology (and related dipole or steric conceptions) has long been common in the teaching of internal rotation barriers [35]. The small barriers to internal twisting about formal single bonds (as in the famous ~3 kcal/mol rotation barrier of ethane) have importance throughout the domain of polymeric interactions, particularly in dictating the folding properties of biopolymers [36] that literally “shape” biological function. Although students gain clear covalent conceptions of the high barriers to twisting about double bonds (involving progressive breaking of the covalent pi-bond), the corresponding energetic penalty for breaking the weaker *hyperconjugative* stabilizations involving valence *anti*bonds was long overlooked or underestimated [37].

Following the examples shown above for H-bonding, we can again easily demonstrate the importance of valence antibonds {σ*_AB_} in molecular shape by merely *deleting* these orbitals from full quantum chemical description and recalculating the *E*^(DEL)^ potential energy surface for the unnatural world in which such essential features of full quantal description are absent.

To that end, we now examine classic examples that exhibit hindered rotation about an internal rotor bond: ethane (CH_3_CH_3_), methylamine (CH_3_NH_2_) and formamide (HCONH_2_). In each case, the torsional bond (C–C, C–N) is of formal “single-bond” character in elementary Lewis-structural description. However, full NRT description of each optimized equilibrium structure reveals bond orders with mysterious small *deviations* from idealized single-bond values, as shown in the panels of Figure 6.

Numerical details of the equilibrium geometries in Figure 6 are summarized in Table 3, showing symmetry-unique values of bond lengths (*R*_ij_, Å), valence angles (*θ*_ijk_, °), and dihedral angles (τ_ijkl_, °).

Aside from more general torsional properties, several interesting questions can be raised concerning the basic equilibrium structural data in Table 3, particularly relating to subtle methyl-group differences between CH_3_CH_3_ and CH_3_NH_2_. In contrast to the *C*_3_ symmetry of the methyl group in ethane, the methyl group in methylamine has the CH(3) bond bent about 6º further off the CN axis than the CH(4) and CH(5) bonds, causing the CH_3_ group to appear “tilted toward the nitrogen lone pair” in the latter case. The τ_HCNH_ twist angle for H(3) is also noticeably different than those for H(4) and H(5), as are the corresponding *R*_CH_ bond lengths in methylamine. All such symmetry-breaking deviations from idealized sp^3^-type geometry seek electronic explanation.

As a first step, we can examine the drastic effect of deleting *all* non-Lewis orbitals, reducing the basis to the doubly occupied Lewis-type NBOs of natural Lewis structure (NLS) description, thus prohibiting any resonance-type conjugative or hyperconjugative contributions from alternative bonding patterns. The panels of Figure 7 show calculated rotational barrier profiles for each species of Figure 6, comparing the relaxed-scan potential energy curve for full (*E*^(full)^; circles, solid line) calculation with the corresponding NLS-deletion curve (*E*^(NLS)^; x’s, dotted line). For HCONH_2_, the *E*(τ_OCNH_) curve is shown only in the range 0–90° (because the low-energy solution switches to the opposite τ_OCNH′_ torsional coordinate beyond that point), but the barrier shapes are shown for twist angles τ in the range 0–180° for other species.

Figure 7 shows that the magnitudes and shapes of rotation barriers vary widely between CH_3_CH_3_ or CH_3_NH_2_ vs. HCONH_2_, indicating distinct “types” of barrier behavior. The *difference* between full and NLS barrier curves is seen to be drastic, despite the fact that the corresponding differences in *total* energy [%-Δ*E* = 100*|(*E*^(NLS)^ − *E*^(full)^)/*E*^(full)^|] are miniscule (<1%). We now seek to identify the most important orbital-level contributions to torsional hindering in each case.

### 4.2. Conjugative (“π*-Type”) Torsional Hindering in HCONH_2_

As famously recognized by Pauling [38], formamide and other amide groups are significantly “stiffened” against torsional deformations by resonance-type conjugation of the O=C–N ↔ O–C=N type. In NBO donor–acceptor language, such resonance mixing is associated with *n*_N_-π*_CO_ delocalization of the nitrogen lone pair (*n*_N_) into the adjacent π*_CO_ valence antibond, as shown in visualizable orbital imagery for HCONH_2_ in Figure 8. From the strong pi-type *n*_N_-π*_CO_ stabilization (and significant NRT bond order; Figure 6) that is achieved in planar alignment of these NBOs, it is evident that attempted out-of-plane twisting must be strongly opposed by the loss of conjugative pi-bonding interaction, analogous to the still stronger opposition to such twisting in CH_2_=CH_2_.

The high barrier (ca. 15 kcal/mol) seen in the right panel of Figure 7 is therefore a rather obvious consequence of *n*_N_-π*_CO_ resonance conjugation, as expressed in the language of NBO donor–acceptor interactions. This is in full accord with Pauling’s original perception, and it serves to confirm (in NBO framework) that the well-known conformational stiffness of amides is the “easiest” type of torsional hindering to understand from elementary resonance-type considerations.

### 4.3. Hyperconjugative (“σ*-type”) Torsional Hindering in CH_3_CH_3_ and CH_3_NH_2_

The remaining panels of Figure 7 feature saturated species in which no pi-bonds, and hence no π*-based donor–acceptor NBO interactions, are present. We now focus on prototypical ethane-type barriers as exemplified by CH_3_CH_3_ and CH_3_NH_2_. These barriers are well known to dictate the thermal and mechanical properties of polymer chains [39].

In each case, one can see that deletion of non-Lewis orbitals practically *obliterates* torsional hindering, leaving residual Δ*E*^(NLS)^(τ) variations (x’s, dotted line) of less than 0.1 kcal/mol. The famous ethane-type rotation barrier (ca. 2.7 kcal/mol) is thereby traced to the miniscule delocalizations of CT-type donor–acceptor interactions, whereas no remnant of the barrier remains with the Lewis-type orbitals of the NLS determinant that describe 99.65% of total electron density and thus provide near-*exact* description of classical-type steric and electrostatic multipole properties of ethane.

By performing more selective deletions, one can verify that σ*-type antibonds (specifically, σ*_CH_) are prerequisite acceptor orbitals, and adjacent σ_C′H′_ bonding NBOs [particularly in trans-like (*antiperiplanar*) orientations] are the corresponding donor orbitals for the *sigma*-type (hyperconjugative) donor–acceptor interactions underlying ethane-type rotation barriers. For this hyperconjugative domain of torsional hindering, the relevant sigma-type NBOs for ethane at equilibrium geometry (analogous to the conjugative pi-type NBOs of Figure 8 for formamide) are displayed in Figure 9, showing the specific σ_C(1)H(3)_-σ*_C(2)H(7)_ interaction and perturbative estimate (*E*_σσ*_^(2)^, kcal/mol) of its strength in staggered (left) and eclipsed (right) conformations. Figure 9 makes clear that each vicinal σ_C(1)H_-σ*_C(2)H′_ interaction is *more* stabilizing (“better overlap”) in a staggered (*anti*, left panel) than eclipsed (*syn*, right panel) conformation, and Figure 7 (left panel) makes clear the devastating loss of significant torsional hindering if such interactions are deleted.

Note that the participating σ_CH_ and σ*_C′H′_ NBOs are each formally “sigma-type” in symmetry and are thus notably different than the pi-type orbitals of Figure 8. Nevertheless, the σ_CH_-σ*_C′H′_ interaction of Figure 9 depicts canted “pi-type” (off-axis) partial bonding character. The NRT bond orders of Figure 6 can therefore be compared in apples-to-apples fashion with those of ethylene and other aliphatic or aromatic species over a *continuous* range of variations.

For methylamine, the various hyperconjugative interactions involving hydride donor NBOs (viz., σ_CH_-σ*_NH_ and σ_NH_-σ*_CH_) are rather similar to those shown in Figure 9 for ethane. Specifically, the *E*_σσ*_^(2)^ stabilizations for σ_CH_-σ*_NH_ interactions are 2.92 and 0.99 kcal/mol in staggered and eclipsed geometries, respectively, while those for σ_NH_-σ*_CH_ interactions are 2.07 and 1.08 kcal/mol, respectively. However, by far the most important new donor–acceptor feature of CH_3_NH_2_ is the nitrogen lone pair (*n*_N_), which interacts strongly with vicinal σ*_CH_ antibonds as depicted in the *n*_N_-σ*_C(2)H(7)_ overlap diagram of Figure 10, with *E*_σσ*_^(2)^ stabilizations of 7.42 and 6.31 kcal/mol in staggered and eclipsed conformations, respectively.

Although the *n*_N_-σ*_C(2)H(7)_ stabilization (7.42 kcal/mol) in equilibrium methylamine (Figure 10) is much stronger than the corresponding σ_CH_-σ*_C′H_ stabilization (2.56 kcal/mol) in ethane (Figure 9), the *difference* between staggered and eclipsed conformations remains greater for ethane (1.57 kcal/mol) than for methylamine (1.11 kcal/mol). Thus, as shown in Figure 7, the overall rotation barrier for methylamine (ca. 1.9 kcal/mol) remains lower than that for ethane (ca. 2.7 kcal/mol).

Nevertheless, the enhanced strength of *n*_N_-σ*_CH_ hyperconjugation in methylamine has pronounced effects on structural features compared to those of ethane. From the structural results summarized in Table 3, we recall that the θ_NCH(7)_ bending angle in staggered CH_3_NH_2_ (ca. 115°) is conspicuously *larger* than the two remaining NCH angles (θNCH(6) and θNCH(8), each ca. 109°) or the corresponding θCCH angles of ethane (ca. 111°). Many other structural, spectroscopic, and reactivity trends reflect the broken symmetry of the methyl group adjacent to a lone pair.

The electronic origin of this curious geometrical anomaly is easily seen in the left panel of Figure 10. From the displayed orbital shapes, one can visually judge that the *n*_N_-σ*_C(2)H(7)_ overlap (and *E*_σσ*_^(2)^) must be further *in*creased if the σ*_C(2)H(7)_ acceptor orbital is tilted *out*ward or the *n*_N_ donor orbital is tilted *in*ward to the CN axis (i.e., by partially “flattening” the amine group). This large θ_NCH_ angular distortion (ca. 6º) from idealized tetrahedral geometry as well as the slight *elongation* (ca. 0.01Å) of the *R*_CH_ bond that lies *anti* to the *n*_N_ lone pair are among the widely recognized stereoelectronic effects [40] of the amine lone pair in directing preferential bond opening, elongation, vibrational red-shift, and reactive β-elimination at the CH bond that lies *anti* to the lone pair in alkyl amines [41]. The necessary and sufficient role of the *n*_N_-σ*_CH_ interaction in all such phenomena can be further verified by $DEL deletion techniques similar to those used above. Related deletion techniques can be used to resolve related configurational issues involving *cis* vs. *trans* isomerization around double bonds [42].

## 5. Orbital-Level Covalency of Steric Repulsions

### 5.1. Orbital-Level Picture of Steric Repulsion

The same orbital-level principles that lead to the *stabilizing* effects of donor–acceptor orbital interactions in hydrogen bonding and rotation barriers can be applied to quantify the *repulsive* effects of donor–donor (filled orbital) interactions in steric phenomena. Just as proper quantum mechanical orbitals for a physical system were described [(cf. discussions surrounding Equations (4a–c)] as behaving like “axes” of a Cartesian coordinate system, so too must the NBOs remain *orthogonal* (“mutually perpendicular”, like Cartesian $\hat{x}$, $\hat{y}$ axes) to conserve dimensionality under rotations.

Starting from the isolated forms of two fully occupied NBOs at infinite separation, one can anticipate that the orbital waveforms must develop additional “ripples” (oscillations of phase from positive to negative sign, with a dividing nodal surface of *zero* amplitude) that ensure preservation of mutual orthogonality as the orbitals are squeezed into the same region of space. The wavy ripples of orthogonal orbitals at finite separation (associated with second derivatives of high curvature) correspond to increased *kinetic energy* of electrons crowded into an ever-smaller spatial volume.

An increase in energy with respect to a decrease in volume corresponds to *pressure* in classical theory, described as “kinetic energy pressure” by Weisskopf [43]. This term emphasizes the strong distinction from familiar potential energy contributions whose dependence on interatomic separation is power-law (e.g., *R*^−1^ for Coulomb potential) rather than the *exponential* (“brick wall”) dependence of steric forces. The steric forces are deeply tied to the *same* quantum principles that govern all covalency-related phenomena, including the *Pauli exclusion principle* that makes explicit the strong electronic aversion to sharing a spatial region with other electrons of the same spin. At a still deeper level, such Pauli-type forces trace to the spooky *anti*symmetry of electronic wavefunctions (change of sign upon exchange of any two electrons) that lacks any counterpart in classical physics.

### 5.2. Steric Repulsions in Dihelium

To see how this works, we first consider the well-known self-aversion of noble-gas atoms in the simple case of two helium atoms. In the long-range limit, the electronic configuration of each He atom can be described as a doubly occupied 1s orbital [(1s_He_)^2^] of spherical symmetry and positive-everywhere phase. Along any axis passing through the nucleus, the orbital *amplitude* peaks in a cusp-like profile at the nucleus, from which it descends symmetrically toward zero in all directions. However, as the distance *R*_He···He_ between the two nuclei diminishes, the outer “wings” of the orbitals begin to overlap, and each orbital begins to develop a small cusp-like feature of *negative* sign (yellow color in surface plots and dashed lines in contour or profile plots) near the opposite nucleus. This additional nodal feature corresponds to the increased kinetic energy that apparently “repels” closer approach of the other center.

Figure 11 portrays these slight deviations from idealized 1s-type symmetry for three *R*_He···He_ approach distances: one (3Å) just beyond the formal van der Waals contact distance (ca. 2.6 Å) [44], another (2Å) where the negative ripple becomes barely visible in the orbital profile and contours, and the final (1Å) where the departures from ideal-1s form are fully evident in the panels displaying 1d profile, 2d contour, and 3d surface plots. The calculated repulsion energy (Δ*E*, kcal/mol) is also given for each interatomic distance, showing the steep increase from negligible (<0.1 kcal/mol), to modest (ca. 1 kcal/mol), to powerful (>90 kcal/mol) in the three successive steps.

In the MO/DFT framework, the STERIC keyword of NBO analysis [45,46] provides simple numerical estimates of steric exchange energy (*E*_SXE_) based on the energy difference between orthogonal and non-orthogonal (“pre-orthogonal” visualization) NBOs. The left panels of Figure 11 include the parenthesized Δ*E*_SXE_(*R*) values (kcal/mol) for each *R*_He···He_ approach distance. The Δ*E*_SXE_ estimates are seen to give qualitatively reasonable agreement with the full potential energy curve throughout the wide range of repulsions for this rare-gas interaction, which by consensus represents the “purest” known example of steric repulsion. Empirical estimates of atom size go back to the van der Waals equation of the 1870s [47], but at that time, there was no proper basis for the simple orbital-based picture of its electronic origin as portrayed in Figure 11. We contend that introductory discussion of steric repulsions can and should be included in the earliest module involving orbital-level conceptions of chemical bonding rather than deferred to a later module with other “noncovalent” left-overs.

### 5.3. Steric Repulsions in cis-2-Butene

Although steric repulsions are most “purely” represented by He···He and other noble-gas interactions, such repulsions can also have significant chemical effects on the energetics, shape, and reactivity of complex molecules. A simple example is *cis*-2-butene, which can be considered a derivative of propene in which a second methyl group is oriented in *cis* configuration at the opposite end of the double bond, as shown in Figure 12. As seen in the left panel, the barrier (Δ*E*_RB_) to methyl-group rotation is about 2 kcal/mol in propene (similar to previous values noted in Section 4), but the corresponding barrier in *cis*-2-butene is significantly *lower* (ca. 1 kcal/mol), as though the presence of the adjacent methyl group serves to “activate” torsional transitions in the latter case. How can this anomaly be explained?

For stereoelectronic reasons similar to those discussed above, a methyl group adjacent to a double bond prefers a conformation with one CH bond *eclipsing* the double bond. As shown in Figure 12, this preference is adopted by both propene and *cis*-2-butene, but in the latter case, the steroelectronic propensity forces “bay-type” coplanarity with unusually close (2.12 Å) distance between co-aligned CH bonds. In accordance with common organic intuition, such a close approach leads to steric clash (rather than attractive “bonding”, as inferred in QTAIM analysis [48]) that *de*stabilizes the eclipsed equilibrium geometry and thereby *reduces* the energetic penalty of conformational transitions. How can this steric effect be quantified with NBO analysis?

For this question, which concerns the repulsions between specific C(1)H(4) and C(9)H(10) bonds, we turn to the second section of STERIC-keyword output. This presents the “pairwise-additive” estimates of steric exchange energy (*E*_SXE_^(pw)^) for each pair of Lewis-type NBOs. The values are displayed in a tabular layout similar to that for second-order *E*_σσ*_^(2)^ donor–acceptor NBO stabilizations, but the listed entries are now “*E*_σσ_^(pw)^” values for donor–donor *repulsions* between highly occupied NBOs, such as the σ_C(1)H(4)_-σ_C(9)H(10)_ NBOs of current interest.

Figure 13 displays the rotation barrier profile [Δ*E*_RB_(τ); circles and solid line] and corresponding *E*_σσ_^(pw)^(τ) repulsions (x’s, dashed line) for the proximal steric contacts (σ_C(1)H(4)_-σ_C(3)H(9)_ in propene and σ_C(1)H(4)_-σ_C(9)H(10)_ in *cis*-2-butene) in each species. Consistent with organic intuition, the *E*_σσ_^(pw)^ repulsion remains relatively negligible (ca. 0.8 kcal/mol at τ = 0° and below the 0.5 kcal/mol computational threshold for τ > 30°) in propene but ranges up to 2 kcal/mol in *cis*-2-butene.

Figure 14 displays NBO imagery of the leading σ_CH_-σ_C′H′_ (donor–donor) repulsion (left) and σ_CH_-σ*_C′H′_ (donor–acceptor) stabilization (right) that contribute to methyl-torsional properties in *cis*-2-butene. Such diagrams, and their comparison with similar orbital diagrams for propene or *trans*-2-butene, make clear how structural bay-type features lead to strong steric effects on chemical properties that must be incorporated in a more comprehensive picture of chemical covalency. Given the high transferability of NBOs and their effectiveness in communicating orbital-level phenomena in a visually intuitive manner, it seems imperative to incorporate steric repulsions as a *corollary* of more unified chemical pedagogy rather than merely another miscellaneous topic in a dichotomized “noncovalent” category.

Finally, we mention that *E*_σσ_^(pw)^ steric repulsions between filled NBOs must always be considered as an offset to the corresponding *E*_σσ*_^(2)^ estimates of donor–acceptor *attractions* to gain a more realistic estimate of net binding energy. In this sense, the steric repulsions cannot be neglected in any comprehensive description of resonance covalency that aims to give a realistic sense of the net energetics of a connected set of NBO interactions.

## 6. Summary and Conclusions

In the present work we have outlined the electronic logic and provided a variety of pedagogical examples that can serve as a foundation for a more enriched and integrated orbital-level formulation of valency and bonding principles for the supramolecular domain. The desired conceptual changes rely on the ready availability of computational tools for accurate orbital-level descriptors and graphical imagery in the modern Wi-Fi classroom or research laboratory. This gives students at all levels convenient access to corresponding descriptors for problems of their own choosing, with confidence that the obtained results are strictly compliant with ongoing research-level advances throughout the molecular and supramolecular sciences.

Expressed concisely, we contend that H-bonding, ethane-type rotation barriers, and related molecular and supramolecular phenomena should not be dismissively labelled or categorized as “noncovalent”, as though dichotomously opposed to the “covalent” principles of molecular bonding that are commonly taught to beginning chemistry students. Such ill-chosen nomenclature tends to distract both beginning and research-level students from the generality and *unity* of basic quantal principles of covalent bonding, including their resonance-type extension to the fractional bond orders of benzene, amides, and other conjugated molecules as well as the analogous sub-integer bond orders of H-bonded and other X-bonded species. For general textual distinctions between intra- and intermolecular bonding phenomena, the language of “molecular vs. supramolecular” (rather than “covalent vs. noncovalent”) is suitable.

Basic reform of chemical conceptions should begin by discounting traditional dipole–dipole rationalizations of hydrogen bonding and other types of supramolecular associations. Instead, students should be taught that *fractional* (resonance-type) covalency extends seamlessly into the *sub*-integer range of bond orders, allowing a fully unified extension of covalent bonding principles into the supramolecular domain. Other conformational and configurational topics can similarly be re-shaped to deprecate VSEPR and other anachronistic rationalizations that typically require “unlearning” as the student progresses [49]. In this manner, the basic concepts of chemistry are broadened to become foundational for an ever-expanding range of biophysical, medical, and materials-related applications, marking a turning point in chemical comprehension that further emphasizes chemistry’s role as the “central science” in current understanding of the natural world.

## Figures and Tables

**Figure 1 molecules-28-03776-f001:**
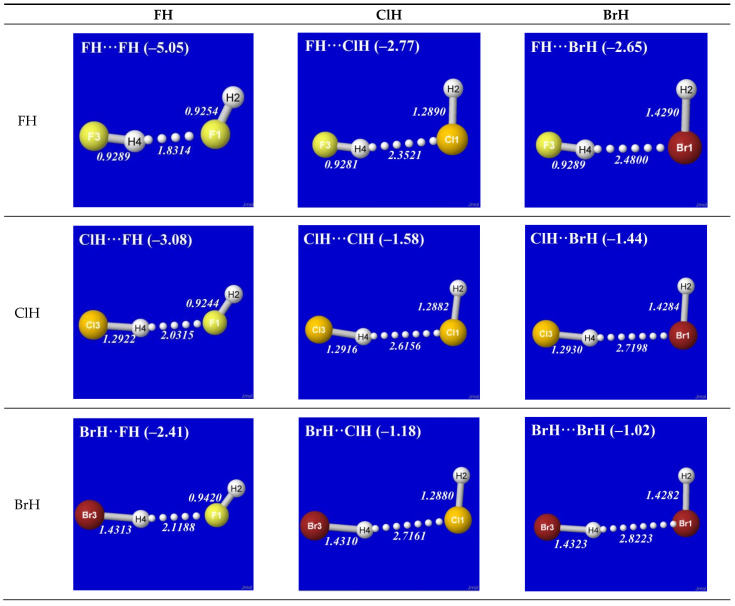
Optimized B3LYP/6-311++G** structures of H-bonded XH···YH (X = F, Cl, Br, H; Y = F, Cl, Br) complexes showing geometrical (italics; Å) and binding energy (parenthesized; kcal/mol) values for each species. Dotted lines denote hydrogen bonding.

**Figure 2 molecules-28-03776-f002:**
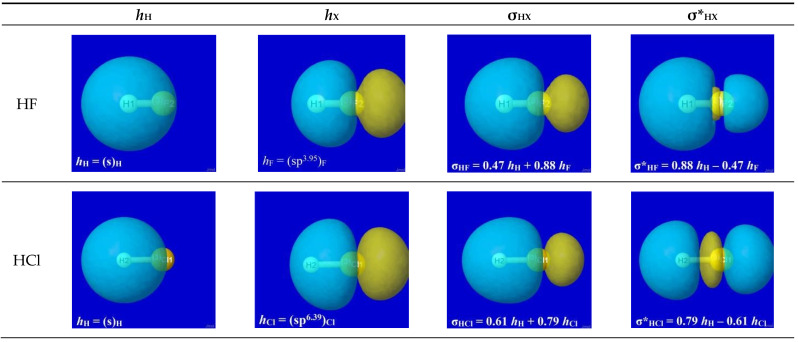
Bonding atomic hybrids *h*_H_ (left panel), *h*_X_ (center panel), and diatomic bonding (P)NBO σ_HX_ (right panel) for each HX monomer (X = F, Cl, Br, H).

**Figure 3 molecules-28-03776-f003:**
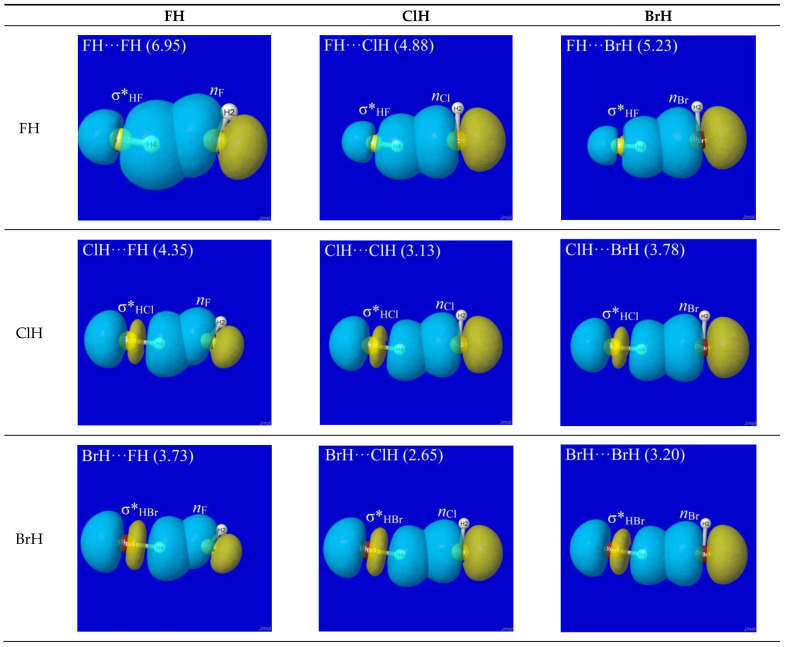
NBO interaction diagrams for leading *n*_Y_-σ*_HX_ interaction of HX···YH complexes (cf. Figure 1) showing the oriented donor lone pair *n*_Y_ and acceptor antibond σ*_HX_ in near-linear alignment that maximizes donor–acceptor stabilization energy (parenthesized *E*_n-σ*_^(2)^; kcal/mol).

**Figure 4 molecules-28-03776-f004:**
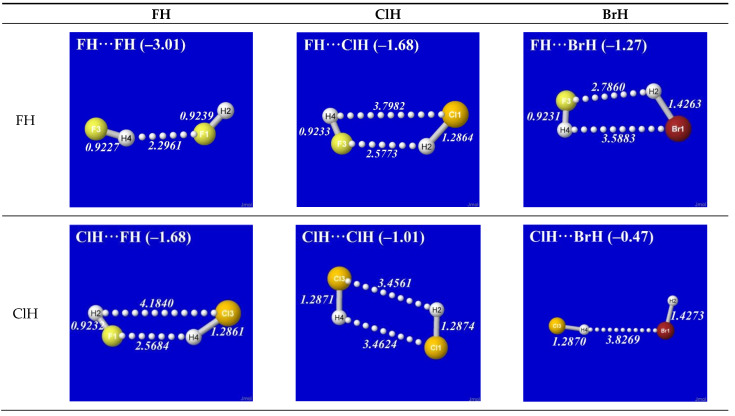
“*E*^(DEL)^” optimizations, similar to those of Figure 1 (with identical initial guess geometry), showing effect of *deleting* hydride antibond NBOs σ*_XH_, σ*_YH_ (and associated charge transfer interactions) with concomitant *loss* of characteristic H-bonding geometry and energetics. Dotted lines serve only to assist structural visualization. (In some cases, the pictured structure is the futile result of incomplete convergence after 100 optimization cycles, which is about four times the default cycle limit).

**Figure 5 molecules-28-03776-f005:**
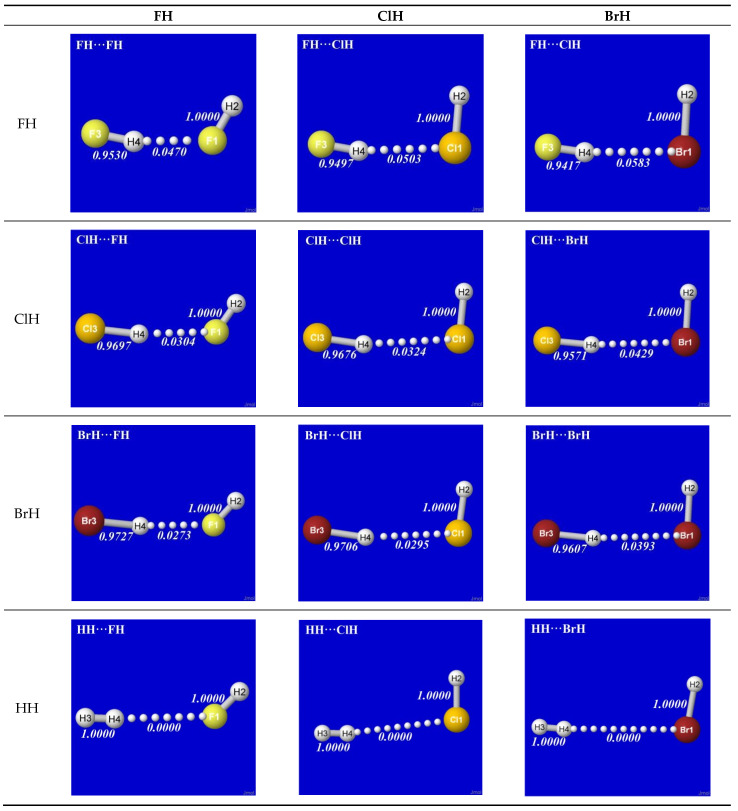
NRT bond orders for XH···YH complexes (cf. Figure 1).

**Figure 6 molecules-28-03776-f006:**
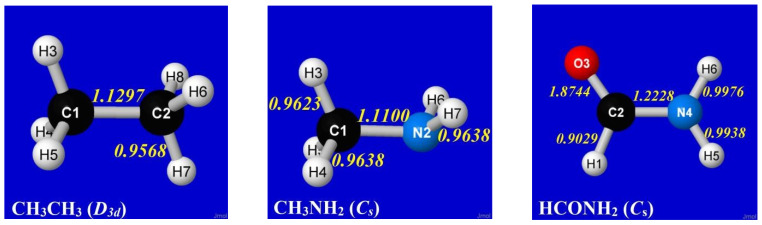
NRT bond orders for equilibrium geometries of ethane, methylamine, and formamide, showing small deviations from unity (nominal “single” bond character) in central torsional bond.

**Figure 7 molecules-28-03776-f007:**
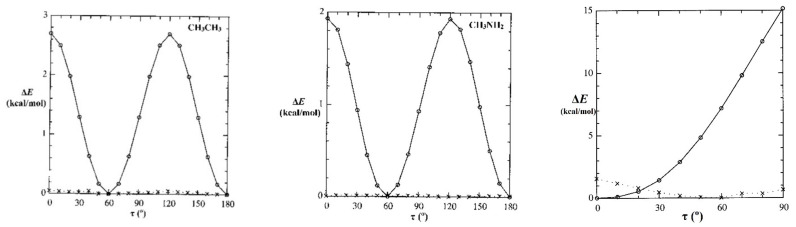
Relaxed-scan *E*(τ) rotation-barrier profiles for the prototype species of Figure 6, comparing the full calculation (*E*^(full)^; circles, solid line) with the NLS counterpart (*E*^(NLS)^; x’s, dotted line) having *all* non-Lewis NBOs deleted (tantamount to removal of all possible resonance-covalency effects).

**Figure 8 molecules-28-03776-f008:**
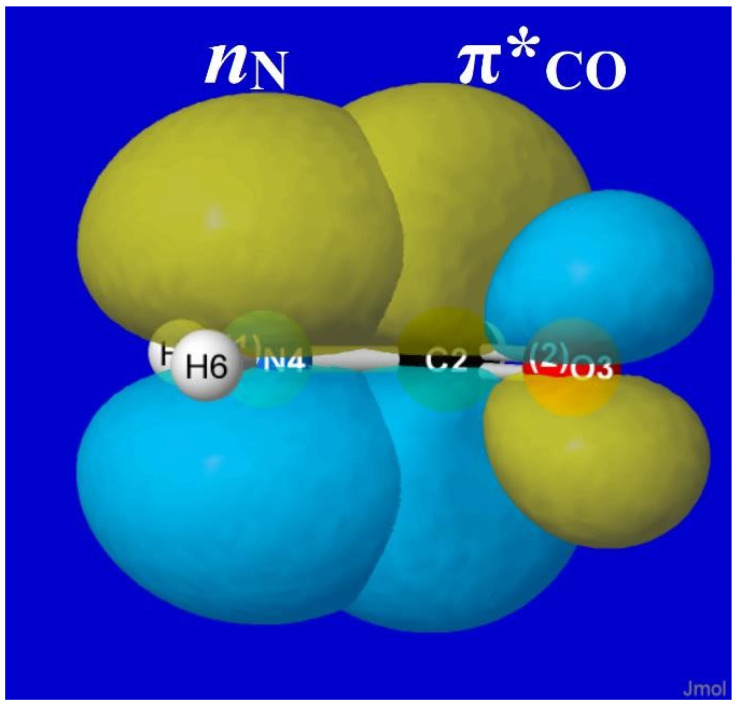
NBO overlap diagram for *n*_N_-π*_CO_ interaction in formamide, depicting the “partial π-bond” resistance to torsional deformation that is the essence of NBO conceptions of barrier origins.

**Figure 9 molecules-28-03776-f009:**
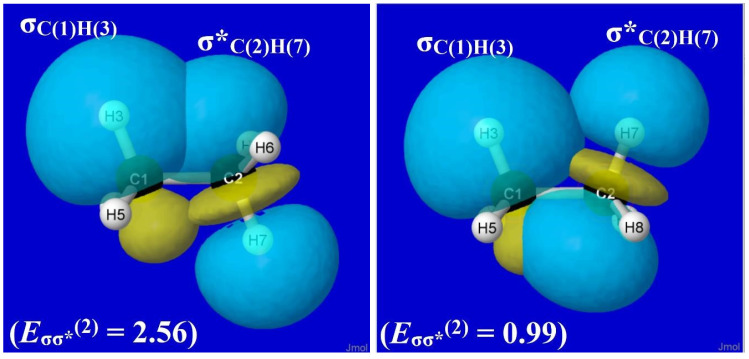
NBO overlap diagrams for staggered (*anti*, left panel) vs. eclipsed (*syn*, right panel) σ_C(1)H(3)_-σ*_C(2)H(7)_ orbital interactions in ethane showing the strengthened stabilization (*E*_σσ*_^(2)^, kcal/mol) in the former case.

**Figure 10 molecules-28-03776-f010:**
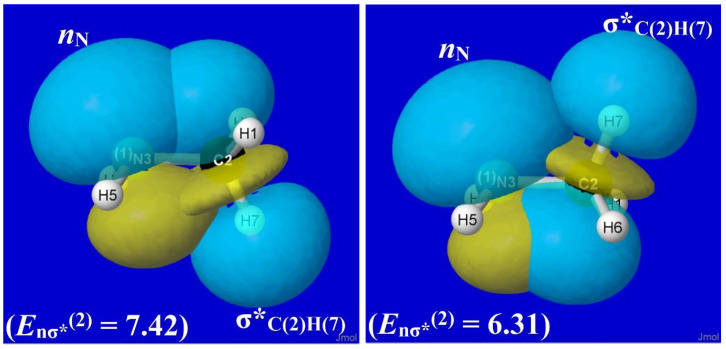
Similar to Figure 9 for the *n*_N_-σ*_C(2)H(7)_ orbital interaction of CH_3_NH_2_.

**Figure 11 molecules-28-03776-f011:**
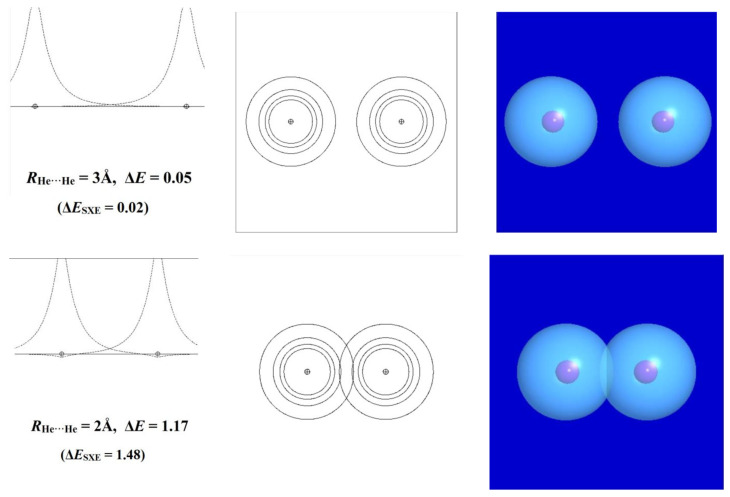
Orbital-level description of He···He steric repulsion for three distances (*R*_He···He_ = 3, 2, and 1Å, top to bottom) showing (left to right) the 1d profile, 2d contour, and 3d surface views of the orbitals and the associated repulsion energy Δ*E* (kcal/mol). Note the phase changes between positive (blue; solid lines) and negative (yellow; dashed lines) values in each panel.

**Figure 12 molecules-28-03776-f012:**
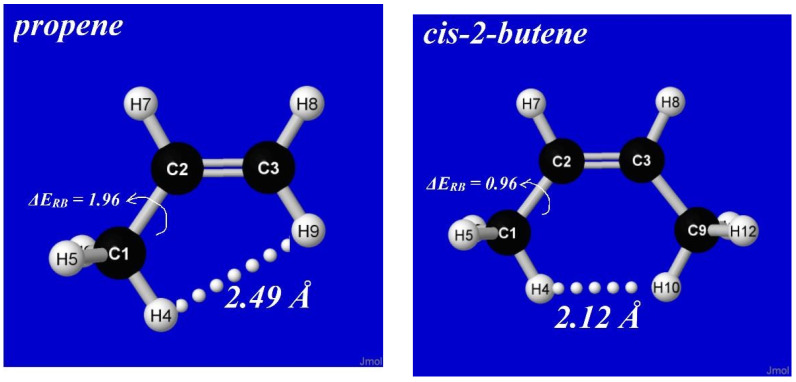
Equilibrium geometry of propene (**left**) and *cis*-2-butene (**right**) comparing methyl rotation barrier (Δ*E*_RB_, kcal/mol) and nearest H···H steric contact distance in each species.

**Figure 13 molecules-28-03776-f013:**
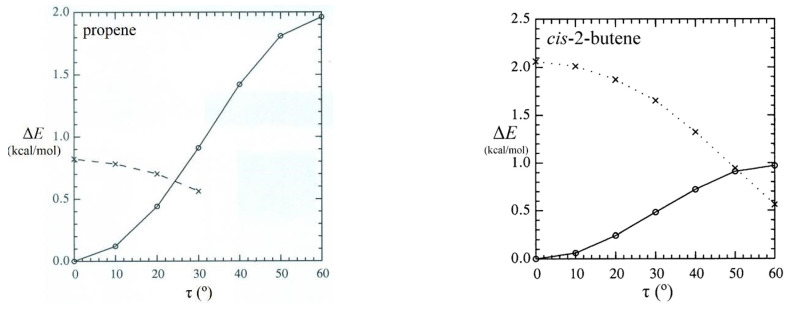
Comparison rotation barriers (circles and solid line) and proximal CH···HC steric repulsions (x’s and dashed line) for propene and *cis*-2-butene showing how increased CH···HC steric repulsion corresponds to reduced torsional barrier in the bay-type geometry of *cis*-2-butene.

**Figure 14 molecules-28-03776-f014:**
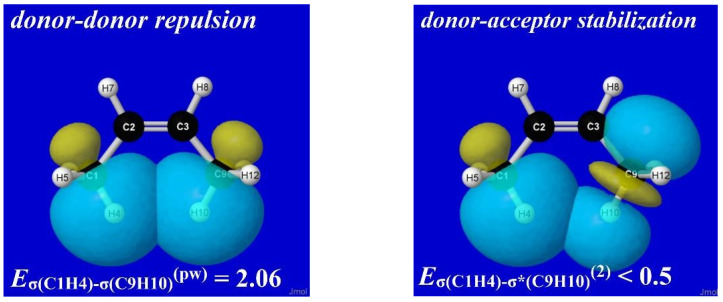
(P)NBO overlap diagrams depicting (**left**) donor–donor repulsions between filled Lewis-type σ_C(1)H(4)_ and σ_C(9)H(10)_ NBOs and (**right**) donor–acceptor stabilizations between Lewis-type σ_C(1)H(4)_ and non-Lewis σ*_C(9)H(10)_ NBOs in *cis-*2-butene.

**Table 1 molecules-28-03776-t001:** Calculated properties of H-bonded XH···YH complexes (cf. Figure 1), showing various experimental observables [binding energy (Δ*E*_HB_; kcal/mol), red-shifted IR frequency (Δν_XH_; cm^−1^), bond lengthening (Δ*R*_XH_; Å)] and theoretical descriptors [NRT bond order reduction (Δb_XH_), NPA intermolecular charge transfer (*Q*_CT_), and second-order perturbative estimate of *n*_Y_-σ*_XH_ stabilization (*E*_nσ*_^(2)^; kcal/mol)].

XH···YH	Δ*E*_HB_	Δν_XH_	Δ*R*_XH_	Δ*b*_XH_	*Q* _CT_	*E* _nσ*_ ^(2)^
HH···BrH	0.00_5_	−4	0.0001	0.0000	0.00075	0.25
HH···ClH	−0.07	−6	0.0003	0.0000	0.00126	0.47
HH···FH	−0.27	−12	0.0007	0.0000	0.00167	1.05
BrH···BrH	−1.02	−61	0.0050	−0.0393	0.01471	3.20
BrH···ClH	−1.18	−43	0.0037	−0.0294	0.01069	2.70
ClH···BrH	−1.44	−87	0.0061	−0.0428	0.01540	3.84
ClH···ClH	−1.58	−64	0.0047	−0.0324	0.01123	3.19
BrH···FH	−2.41	−32	0.0040	−0.0273	0.00927	4.07
FH···BrH	−2.65	−162	0.0067	−0.0583	0.01676	5.28
FH···ClH	−2.77	−140	0.0059	−0.0503	0.01424	4.95
ClH···FH	−3.08	−56	0.0053	−0.0303	0.00977	4.76
FH···FH	−5.05	−139	0.0067	−0.0470	0.01231	7.18

**Table 2 molecules-28-03776-t002:** Pearson correlation coefficients (χ) for each pair of calculated properties (Table 1) of XH···YH complexes (cf. Figure 1).

χ	Δ*E*_HB_	Δν_AH_	Δ*R*_AH_	Δ*b*_AH_	*Q* _CT_	*E* _nσ*_ ^(2)^
Δ*E*_HB_	1.0000	0.7534	−0.7961	−0.7260	−0.5952	−0.9604
ΔνAH	0.7534	1.0000	−0.8680	0.9167	−0.8181	−0.8527
Δ*R*_AH_	−0.7961	−0.8680	1.0000	−0.9734	0.9523	0.9274
Δ*b*_AH_	−0.7260	0.9167	−0.9734	1.0000	−0.9729	−0.8792
*Q*CT	−0.5952	−0.8181	0.9523	−0.9729	1.0000	0.7901
Δ*E*_nσ*_^(2)^	0.9604	0.8527	−0.9274	0.8792	−0.7901	1.0000

**Table 3 molecules-28-03776-t003:** Optimized geometrical parameters for symmetry-unique bond lengths (*R*_ij_), valence angles (*θ*_ijk_), and dihedral twist angles (τ_ijkl_) of CH_3_CH_3_, CH_3_NH_2_, and HCONH_2_ (cf. Figure 6).

	*R*_ij_ (Å)	*θ*_ijk_ (°)	τ_ijkl_ (°)
CH_3_CH_3_	*R*_C(1)C(2)_ (1.5307) *R*_C(1)H(3)_ (1.0937)	*θ*_H(3)C(1)C(2)_ (111.36)	*τ*_H(3)C(2)C(3)H(6)_ (60.00)
CH_3_NH_2_	*R*_C(1)N(2)_ (1.4660) *R*_C(1)H(3)_ (1.1003) *R*_C(1)H(4)_ (1.0927) *R*_N(2)H(6)_ (1.0144)	*θ*_H(3)C(1)N(2)_ (115.13) *θ*_H(4)C(1)N(2)_ (109.17) *θ*_H(7)N(2)C(1)_ (111.14)	*τ*_H(3)C(1)N(2)H(7)_ (59.61) *τ*_H(4)C(1)C(2)H(7)_ (−61.97)
HCONH_2_	*R*_H(1)C(2)_ (1.1060) *R*_C(2)O(3)_ (1.2117) *R*_C(2)N(4)_ (1.3608) *R*_N(4)H(5)_ (1.1066) *R*_N(4)H(6)_ (1.0091)	*θ*_H(1)C(2)N(4)_ (112.38) *θ*_O(3)C(2)N(4)_ (124.93) *θ*_C(2)N(4)H(5)_ (121.39) *θ*_C(2)N(4)H(6)_ (119.48)	*τ*_O(3)C(2)N(4)H(6)_ (0.00)

## Data Availability

Not applicable.

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
