# Peer review of "“Noncovalent Interaction”: A Chemical Misnomer That Inhibits Proper Understanding of Hydrogen Bonding, Rotation Barriers, and Other Topics"

_molecules, 2023, doi:10.3390/molecules28093776_

Round 1

Reviewer 1 Report

The title of this paper, "Noncovalent Interaction”: A Chemical Misnomer that Inhibits Proper Understanding of Hydrogen Bonding, Rotation Barriers, and Other Topics", the Abstract and the Summary and Conclusions sections of this paper are in my opinion not conveying the same information.  I recommend that the author modify his title to reflect better the content of the paper.

The author does not give a balanced view of what he calls “neat, simple, and wrong” [7].  I suggest that the author look at the following papers:

PCCP, 20, 30076 (2018); Struct. Chem. 30, 1153 (2019); Crystals 10, 76 (2020); ChemPhysChem 22, 1201 (2021).

There is a continuum in chemical bonding that ranges from very weak to very strong.  These include what some call noncovalent and covalent.  The same forces are involved in both.  As interactions become stronger the effect of polarization increases.

I recommend that the author modify his title and manuscript to better describe and fully discuss his topic.

Author Response

Comments and Suggestions for Authors

The title of this paper, "Noncovalent Interaction”: A Chemical Misnomer that Inhibits Proper Understanding of Hydrogen Bonding, Rotation Barriers, and Other Topics", the Abstract and the Summary and Conclusions sections of this paper are in my opinion not conveying the same information.  I recommend that the author modify his title to reflect better the content of the paper.

I have not altered the title, but instead inserted a new penultimate paragraph in the Summary and Conclusions section (p. 32) to better link it with the title in “conveying the same information”.

The author does not give a balanced view of what he calls “neat, simple, and wrong” [7].  I suggest that the author look at the following papers:

PCCP, 20, 30076 (2018); Struct. Chem. 30, 1153 (2019); Crystals 10, 76 (2020); ChemPhysChem 22, 1201 (2021).

There is a continuum in chemical bonding that ranges from very weak to very strong.  These include what some call noncovalent and covalent.  The same forces are involved in both.  As interactions become stronger the effect of polarization increases.

I recommend that the author modify his title and manuscript to better describe and fully discuss his topic.

I have now revised the paper to include a new Section 3.5 (pp. 17-19) that includes the requested references [29-32] (as well as supporting [33,34]) and discusses the principal points of agreement and disagreement of the “Sigma-Hole” Picture of H/X-Bonding” with the perspective of this work.  This major revision in turn required renumbering ensuing references throughout the remainder of the work.

Reviewer 2 Report

This manuscript aims at a presentation of the capabalities of NBO concerning its analysis of H-bonding and how it should be assessed. The authors in this work made the effort to write it in a very educational way trying to demystify how hard this may seem at times, but without actually being. And it reaches its goals in my opinion, with the author stating that the aim of this review is mostly educational. And it is!, alongside the excellent level of the SI material with working examples illustrating the discussion. 

Wrapping up, this is something I would use in my lectures and it is welcome as NBO is a very nice and simple concept of how theory can measured and visualized.

My very minor comments are: 

1. The "tricky dog" hyperlink works and could also hold the proper link referenced.

2. In page 4,, when introducing Figure 2, the author mentions "3d shapes", which could be misleadign to an orbital. please correct to "3D".

Despite the above suggestions, this manuscript can be published as is.

Author Response

Comments and Suggestions for Authors

This manuscript aims at a presentation of the capabalities of NBO concerning its analysis of H-bonding and how it should be assessed. The authors in this work made the effort to write it in a very educational way trying to demystify how hard this may seem at times, but without actually being. And it reaches its goals in my opinion, with the author stating that the aim of this review is mostly educational. And it is!, alongside the excellent level of the SI material with working examples illustrating the discussion. 

Wrapping up, this is something I would use in my lectures and it is welcome as NBO is a very nice and simple concept of how theory can measured and visualized.

I thank the Reviewer for this assessment.

My very minor comments are: 

  1. The "tricky dog" hyperlink works and could also hold the proper link referenced.
  2. Page 3 is now amended to read “‘tricky dog magnets (which Google)” to avoid a long-winded commercial link insertion, yet provide helpful guidance to a print reader.
  3.  
  4. In page 4,, when introducing Figure 2, the author mentions "3d shapes", which could be misleadign to an orbital. please correct to "3D".

 Corrected (with thanks).

Despite the above suggestions, this manuscript can be published as is.

Reviewer 3 Report

This work emphasizes that the covalent contribution to "noncovalent" interactions might play a dominant role in the case of the formation of hydrogen bonds or, in particular, during rotation around single bonds. This work evokes very ambivalent feelings. On the one hand, it is exceptionally well-written and the author describes the results of quantum chemical calculations in a simple and understandable form. But, on the other hand, the described concept is very well-known. In particular, the author recommend to use this orbital concept to describe "noncovalent" interactions for teaching students, but in fact all these ideas were tought to us in the basic course of Organic Chemistry and further in the course of Stereochemistry independently by two different professors when I was a student  (15 years ago; and it seems to me that they use this concept many years before me studying in university). Maybe this is a feature of our institute and this concept is not used in major part of other universities. If so, I recommend to publish this work in present form (after correction of a typo, see below). But before this, I strongly recommend to send this manuscript to a set of academic reviewers from different countries to check an abundance of this concept among worldwide studying process.

The typo: section 3.1, first line — "cRoss-interactions".

Author Response

This work emphasizes that the covalent contribution to "noncovalent" interactions might play a dominant role in the case of the formation of hydrogen bonds or, in particular, during rotation around single bonds. This work evokes very ambivalent feelings. On the one hand, it is exceptionally well-written and the author describes the results of quantum chemical calculations in a simple and understandable form. But, on the other hand, the described concept is very well-known. In particular, the author recommend to use this orbital concept to describe "noncovalent" interactions for teaching students, but in fact all these ideas were tought to us in the basic course of Organic Chemistry and further in the course of Stereochemistry independently by two different professors when I was a student  (15 years ago; and it seems to me that they use this concept many years before me studying in university). Maybe this is a feature of our institute and this concept is not used in major part of other universities. If so, I recommend to publish this work in present form (after correction of a typo, see below). But before this, I strongly recommend to send this manuscript to a set of academic reviewers from different countries to check an abundance of this concept among worldwide studying process.

I’m happy to hear that proper hyperconjugative teaching about rotation barriers was well established in courses at your university, but I believe that beginning classes elsewhere tend not to be so fortunate (particularly with respect to the fractional bond orders of H-bonding).  Reviewers of this work probably represent such a variety of countries.

The typo: section 3.1, first line — "cRoss-interactions".

 Corrected (with thanks).

Round 2

Reviewer 3 Report

Although I still believe that this work describes well-known concept, I recommend to accept it in the present form, because the material of this manuscript represents well-organized discussion of covalent contribution to "noncovalent" interactions, which might be useful for a range of chemists.